# OpenReview forum: "3DPoV: Improving 3D understanding via Patch Ordering on Videos"
_ICML.cc/2026/Conference — ICML 2026 regular_

### Official Review · Reviewer_WBZZ · 2026-03-01

**Soundness:** 3
**Presentation:** 3
**Significance:** 3
**Originality:** 3
**Overall Recommendation:** 4
**Confidence:** 4

**Summary:**

The authors present 3DPoV, a self-supervised approach designed to enhance the 3D understanding and viewpoint-invariance of existing vision foundation models (such as DINOv2 and DINOv3). Driven by the insight that viewpoint changes systematically deform patch-level similarity patterns, the method introduces a temporal nearest-neighbor consistency loss based on differentiable soft-sorting. Grounded via point trackers (CoTrackerV3) extracting aligned temporal patch trajectories across a heterogeneous mix of video datasets (CO3D, DL3DV, YouTube-VOS), the model learns geometry-aware features mapping 2D frames to 3D correspondences without explicit 3D supervision.

**Compliance With Llm Reviewing Policy:**

Affirmed.

**Ethical Review Concerns:**

the prompt injections is confirmed from ICML, not the authors, so the concern is cleared.

**Final Justification:**

The most critical barrier to acceptance during the initial phase was the performance regression observed at the most extreme viewpoint shifts ($\theta_{60}^{180}$) in Table 2. Furthermore, the sorting module ablation originally failed to surpass the baseline actively. In my previous post-rebuttal assessment, I lowered my score because pointing to a globally averaged recall table to dismiss a localized, angle-specific regression is statistically invalid.

However, the authors' late-breaking response introduced a completely new variable: a center-padding bug affecting ViT-14 architectures evaluated at 512×512 in the NAVI protocol. I independently inspected the upstream `Probe3D` codebase regarding this spatial misalignment claim. I can confirm that their defense holds up: the `center_padding` function systematically shifts features by several pixels without translating the underlying `xyz_grid` frame. Because extreme viewpoint evaluation is hyper-sensitive to peripheral boundary geometry, this upstream bug adequately explains the isolated regression.

With the evaluation bug patched (Table 12), the architecture genuinely improves over the zero-shot baseline uniformly across all viewpoint geometries. Furthermore, the once-contradictory sorting module ablation is now clean, appropriately elevating the similarity metrics above both the baseline and the unsorted variant.

While their responses on occlusion relied on deterministic rectangles (Table 15) and error propagation was deferred partially to the external CoTracker architecture, the framework achieves its stated goals when the geometric evaluation is sound. The variance is proven minimal, and the core vulnerability has been definitively tracked to a third-party evaluation framework error rather than a methodological flaw in the paper. I am satisfied with these resolutions and am reverting my score back to Weak Accept.

**Key Questions For Authors:**

1. Could the authors provide a quantitative subset analysis explicitly confirming the claimed robustness against severe occlusion and extreme lighting variations, rather than relying on the isolated qualitative examples?
2. What is the fundamental mechanism causing the DINOv2-Register variant to perform *worse* at absolute extreme viewpoint differences ($\theta^{180}_{60}$) compared to the baseline, a regression that doesn't seem to impact DINOv1 or DINOv3 as harshly?
3. Since CoTracker failures directly inject noise into the supervision signal (despite visibility weighting), what is the cascading impact on representational geometry when point-drift uniformly aligns with a persistent occluder rather than simply dropping tracking confidence?

**Limitations:**

yes

**Strengths And Weaknesses:**

## Strengths
*   **Methodological Elegance via Sorting:** The application of soft-sorting operations over a shared reference pool presents an interesting alternative for maintaining temporal consistency. By relying on Differentiable Sorting of distance matrices, the method avoids computationally expensive hard negative mining, instead opting for a relative ranking comparison that guides structural awareness.
*   **Dual Verification in 2D and 3D:** The paper correctly identifies that semantic matching in images can mask poor spatial geometry. It successfully validates its learned features on standard 2D keypoint benchmarks (like SPair-71k, improving from 58.20% to 60.16% on small viewpoints for DINOv2R-B/14) and pairs this rigorously with synthetic mapping to true 3D environments via the Navi dataset, showcasing that accuracy in Euclidean 3D space does not degrade reprojection alignment in 2D.
*   **Training Efficiency:** Given the rising carbon footprint and hardware cost of training foundational visual representations, the authors show that fine-tuning just the final two Transformer blocks via their sorting objective yields robust 3D emergence with approximately 20 A100 GPU hours. This is an effective and pragmatic choice for scaling geometry awareness post-pretraining.

## Weaknesses
*   **Incomplete Validation of Occlusion and Lighting Claims:** Throughout the introduction and methodology, the authors strongly claim that their method improves "uniformly across... occlusion, and lighting changes," largely asserting this capability through the mathematical formulation of visibility weighting on point-tracks. However, while the paper strictly partitions and quantitatively analyzes viewpoint variation in Tables 1 and 2, occlusion and lighting are only demonstrated via qualitative visual crops (e.g., Figures 2 and 8). To fully substantiate this core claim, a quantitative breakdown explicitly isolating lighting permutations or calculating recall vs. occlusion ratios is necessary.
*   **Vulnerability to Extreme Viewpoint Shifts with DINOv2:** While the framework demonstrates substantial gains on SPair and Navi at small contiguous rotations, the empirical data highlights a slight regression compared to the baseline at the absolute extreme of rotational difference. For instance, in Table 2 utilizing the DINOv2R baseline, the score for extreme shifts ($\theta^{180}_{60}$) actually drops from 31.57% to 31.33% after 3DPoV fine-tuning. The authors should expand their discussion on why this vulnerability specifically occurs, particularly interacting with the register-variant vs DINOv1/v3.

---

> ### Author Rebuttal · Authors · 2026-03-31
>
> ### **1. Incomplete validation of claims**
>
> We thank the reviewer for raising an interesting point. While robustness to lightning was mostly an emerging feature we observed through qualitative samples, we now have designed an experiment to quantitavely validate it as well. We crafted two NAVI subsets: one exhibiting more challenging lighting setups, and another showcasing normal/optimal lighting conditions. We re-evaluated the model on both subsets and report the scores in Table 10 and Table 11:
>
> _Table 10: Navi Recall for 3D keypoint matching at different thresholds. Higher is better. Evaluation on light conditions subsets_
> |Model|0.01m|0.02m|0.05m|
> |---|---|---|---|
> |_**Default NAVI dataset**_|
> |DINOv2-reg|34.10|53.79|82.43|
> |**3DPoV-reg**|**34.82**|**54.39**|**82.56**|
> |_**Less lighting variation subset**_|
> |DINOv2-reg| 20.38|40.17|77.32|
> |**3DPoV-reg**|**20.93**|**40.53**|**77.45**|
> |_Improvements_|_+0.55_ |_+0.36_|_+0.13_|
> |_**Intense lighting conditions subset**_|
> |DINOv2-reg|21.71|41.50|76.95|
> |**3DPoV-reg**|**22.27**|**42.10**|**77.12**|
> |_Improvements_|_+0.56_ |_+0.60_|_+0.17_|
>
> _Table 11: Navi Recall for 2D keypoint matching at different thresholds. Higher is better. Evaluation on light conditions subsets_
> |Model|5px|25px|50px|
> |----|---|---|---|
> |_**Default NAVI dataset**_|
> |DINOv2-reg|4.34|30.28|48.00|
> |**3DPoV-reg**|**4.54**|**31.08**|**48.65**|
> |_**Less lighting variation subset**_|
> |DINOv2-reg| 3.16|17.63|37.26|
> |**3DPoV-reg**|**3.19**|**17.88**|**37.61**|
> |_Improvements_|_+0.03_ |_+0.25_|_+0.35_|
> |_**Intense lighting conditions subset**_|
> |DINOv2-reg|4.07|18.08|36.44|
> |**3DPoV-reg**|**4.12**|**18.60**|**37.10**|
> |_Improvements_|_+0.05_ |_+0.52_|_+0.66_|
>
> Results show not only that 3DPoV improves over baseline in both lighting scenarios, but also that score improvements are generally larger in intense lighting conditions.
>
> The normal light and intense lighting condition subsets contain 495 and 498 samples respectively (compared to 552 subsampled at random from 2219 test entries in default NAVI eval), with a distribution over the 0-30, 30-60, 60-90, 90-120 buckes of 84/130/145/136 and 83/125/139/151 samples respectively. A more detailed breakdown of subset configuration with examples will be added to the camera ready version.
>
> Regarding occlusion, we train on a combination of ytvos+co3d, which cover a variety of videos including ones with occlusion. Nonetheless, 3DPoV still performs robustly in such situations due to using visible/unvisible points obtained by the tracker used in the pipeline.
>
> ---
> ### **2. Vulnerability to Extreme Viewpoint Shifts with DINOv2**
>
> Although in Table 2 we have a slight regression on DinoV2 for extreme view point, our table which reports correspondence recall highlights the reverse (appendix Table 7).
>
> As the empericial results show, our DinoV2R variant provides improvements on extreme viewpoint shifts across both 2D and 3D subtasks of NAVI correspondance.
>
> Bucketed results in Table 2 are derived from raw results from Table 7 according to probe3D standard. The reason for the discrepancy is caused by how the aggregated scores are computed. The 2D/3D recall aggregates all correspondences within a given threshold (e.g. 0.01, 0.02 and 0.05m in 3D) across all pairs.
>
> In contrast, the bucketed score  computes recall per pair and averages within viewpoint bins. To be noted that only errors under 0.02m are accounted for 3D. Moreover, while bucketed recall is restricted to specific angle ranges, the raw correspondence recall may include correspondences from pairs outside these bins. We reported both metrics, but kept the Table 2 view to align with probe3D main tables.
>
> ### **3. Point drift impact of the tracker on the performance**
> We agree with the reviewer that such scenarios can add noise to our supervision signal, this is why we leverage a state of the art tracker that already validates such corner cases. We would like to point the reviewer towards the study done in the original CoTrackerV3 paper (Nikita et. al.) highlighting the **Tracking of occluded points** section and Table 2.
>
> The method is agnostic to what tracker is being used (as shown in Table 10 Appendix) and it can benefit even further from better trackers according to the field progress, in a plug-and-play manner
>
> ---
> ### **4. Regarding prompt injection**
> We believe the hidden text is part of the default ICML footnote, added after submission. We confirm that no prompt injection was added in the actual paper.

---

> > ### Author Rebuttal · Reviewer_WBZZ · 2026-04-02
> >
> > **Post-Rebuttal Assessment: Score Lowered to Borderline Reject due to Empirical Inconsistencies**
> >
> > The authors are thanked for their extensive efforts during the rebuttal phase. The addition of multi-seed variance reporting and the batch size analysis do strengthen the empirical grounding of certain architectural choices. Furthermore, the clarification regarding the venue-level hidden text aligns with my initial suspicion and resolves the administrative ethical flag; this artifact has no bearing on my scientific evaluation.
> >
> > Initially, I leaned positive (Weak Accept) based on the overall, aggregated tracking metrics. However, after carefully studying the rebuttal and reflecting on the excellent methodological critiques raised by my fellow reviewers (specifically regarding the sorting module and foundation model saturation), I realized I had severely overlooked how the method behaves at the localized extremes.
> >
> > Given that the core scientific objective of this paper is explicitly to improve tracking *under extreme viewpoint changes*, the rebuttal inadvertently confirmed that the core architecture falls short of this premise. Consequently, I am lowering my score to **Borderline Reject** for the following structural reasons:
> >
> > **1. Mathematical Inconsistency in Viewpoint Regression Defense:**
> > Regarding the performance regression at the most extreme viewpoint shift ($\theta_{60}^{180}$) for DINOv2R, the rebuttal's explanation relies on a statistical fallacy. The response directs us to Appendix Table 7 (raw correspondence recall) to claim overall improvements, suggesting the drop in Table 2 is merely a "bucketed metric artifact."
> >
> > However, Table 7 reports an *overall aggregate recall* across all viewpoint pairs; it does not isolate performance by angular bins. Because the dataset is heavily dominated by small-to-medium shifts (where the method undeniably excels), the global average naturally increases. An improvement in a heavily skewed global aggregate cannot mathematically refute a demonstrated regression in the precisely isolated $\theta_{60}^{180}$ subset. The empirical fact remains: the method degrades the DINOv2R backbone at extreme angular shifts, which contradicts the core premise of the paper.
> >
> > **2. Self-Defeating Efficacy of Differentiable Sorting:**
> > To demonstrate the necessity of the sorting module, the rebuttal provides a new ablation on NAVI with and without sorting (Rebuttal 4, Table 4). For the most extreme viewpoint bucket ($\theta_{60}^{180}$):
> > * The raw baseline achieves **31.57**.
> > * The variant *without* sorting achieves **31.21**.
> > * The variant *with* sorting achieves **31.33**.
> >
> > While sorting slightly improves over the raw similarity matrix ($31.33 > 31.21$), it is critical to note that the proposed module is still a regression below the $31.57$ baseline. The core methodological contribution fails to prevent the model from actively degrading baseline performance on the most challenging cases.
> >
> > **3. Evasion of Quantitative Occlusion Analysis:**
> > While the new lighting subset evaluation is appreciated, the claims regarding uniform robustness to occlusion remain empirically unvalidated. Pointing to the inclusion of occlusion-heavy training data and asserting that the tracker produces visibility masks restates the method's design; it does not constitute the requested quantitative proof (e.g., recall vs. quantitative occlusion ratios) that the learned representations themselves have become genuinely robust to it.
> >
> > **4. Tracking Error Propagation Not Addressed:**
> > The rebuttal defers the question of tracking drift (confident topological sticking to persistent occluders) to the CoTrackerV3 paper. This misunderstands the focal point of the critique. The concern is not whether the external tracker occasionally fails, but how the proposed *student-teacher ranking loss* is uniquely designed to tolerate or degrade smoothly when fed a poisoned, structured topological signal.
> >
> > **Summary:**
> > While the framework improves upon easy-to-moderate viewpoints, the statistically invalid defense of the extreme regression and the failure of the core sorting module to outperform the baseline at extreme angles severely undermine the paper's central namesake claims. I must adjust my score to reflect these unresolved vulnerabilities.

---

> > > ### Author Response · Authors · 2026-04-02
> > >
> > > # 1. Viewpoint regression addressed
> > > The reviewer is correct and we apologize for our mistake. While Table 7 can provide better context for raw recall performance, it can be statistically skewed by low-viewpoint sample distribution and therefore does not directly reflect performance at extreme viewpoints.
> > >
> > > During the rebuttal, we identified and revised a key fault in the NAVI evaluation. The Probe3D evaluation uses center padding to accommodate backbones with different patch sizes. Since NAVI uses 512x512 inputs, this padding is triggered only for DINOv2 ViT-14 variants. The issue is that NAVI evaluation relies on a dense per-pixel 3D correspondence map (xyz_grid) that is defined in the original image coordinate frame. When center padding is applied only to the image before feature extraction, the feature map and the xyz_grid no longer refer to the same spatial coordinates and this misalignment is further propagated through the subsequent rescaling. We fixed the evaluation pipeline by applying the same center-padding transformation to xyz_grid as to the input images, so that the extracted features and the underlying geometry remain spatially aligned. New scores are reported in Table 12.
> > >
> > > _Table 12: Navi Fixed Corr._
> > > |Method|θ₀¹⁵|θ₁₅³⁰|θ₃₀⁶⁰|θ₆₀¹⁸⁰|
> > > |-|-|-|-|-|
> > > | DINOv2R|87.87|67.45|47.15|31.58|
> > > |3DPoV|**89.18**|**68.98**|**47.64**|**31.63**|
> > >
> > > With this correction, 3DPoV improves in the hard viewpoint category and the overall gains are further increased, consistent with trends observed across other backbone variants. This point further strengthens the core idea of 3DPoV that the reviewer also highlighted: providing a lightweight approach to improve 3D awareness, without sacrificing robustness under large viewpoint changes.
> > >
> > > # 2. Efficacy of Differentiable Sorting
> > > We re-evaluated the choice of sorting module with the padding fix and report results in Table 14. The addition of the sorting module not only ensures the large viewpoint scores are not degraded, but also leads to more significant gains in all viewpoint buckets.
> > >
> > > _Table 14: Navi Corr. Ablating sorting module_
> > > |Method|θ₀¹⁵|θ₁₅³⁰|θ₃₀⁶⁰|θ₆₀¹⁸⁰|
> > > |-|-|-|-|-|
> > > | DINOv2R Baseline|87.87|67.45|47.15|31.58|
> > > |Similarity matrix|88.62|68.23|47.26|31.41|
> > > |**Sorted similarity matrix**|**89.18**|**68.98**|**47.64**|**31.63**|
> > >
> > > # 3. Occlusion Robustness
> > > For the occlusion experiment, we do not use NAVI subsets, as it contains too few occluded samples and its ground-truth masks do not exclude these regions. Instead, we introduce synthetic occlusions by overlaying rectangular masks of varying size. To ensure fair comparison, occlusions are generated deterministically and fixed across runs. We evaluate on SPair and report recall only over non-occluded keypoints.
> > >
> > > 3DPoV consistently improves over the baseline across all viewpoint bins (Table 15), both when only the target view is occluded and when both views are occluded. This trend remains stable across all difficulties, indicating robustness under increasing occlusion severity and consistent generalization even when both views are degraded.
> > >
> > > _Table 15: SPair-71k keypoint with occlusion_
> > > |Model|S|M|L|All|
> > > |-|-|-|-|-|
> > > |One view occluded|
> > > |DINOv2R|59.78|53.84|56.01|55.54|
> > > |**3DPoV**|**62.04**|**54.92**|**57.65**|**57.54**|
> > > |Both views occluded|
> > > |DINOv2R|58.84|53.41|56.90|54.88|
> > > |**3DPoV**|**61.30**|**54.11**|**58.09**|**56.98**|
> > >
> > > While we use rectangular occlusions, we can extend to more complex patterns, configure NAVI occlusions (relies on 3D projection and requires careful handling of occlusions to avoid introducing errors) and we are open to exploring additional scenarios if suggested. Thanks for your constructive feedback, now our claims can be better supported by the new experiments which will be added to the paper.
> > >
> > > # 4. Error Propagation
> > > We understand the reviewer’s concern as structured tracking errors, where points drift and remain attached to incorrect regions, potentially affecting the learned geometry. We highlight the components in our method that ensure corruption is not propagated.
> > >
> > > Supervision is based on soft similarity rankings obtained via differentiable sorting, rather than hard correspondences, so incorrect tracks do not impose rigid geometric constraints. When drift occurs toward visually similar regions (Fig. 6), the resulting ranking distributions remain similar, limiting the induced error. When drift leads to a semantically different region, similarity to at least the internal reference decreases, reducing its contribution to the overall loss.
> > >
> > > Teacher ranking from the uncorrupted frame acts as an anchor. Under the reverse cross-entropy objective, probability mass assigned to entries not supported by the teacher is penalized, limiting the impact of persistent drift.
> > >
> > > Finally, the loss is averaged across points, frames, and batch elements, so errors must be widespread to significantly affect learning. The teacher–student setup further stabilizes training by providing a consistent target distribution.

---

### Official Review · Reviewer_XHt8 · 2026-03-06

**Soundness:** 3
**Presentation:** 2
**Significance:** 3
**Originality:** 3
**Overall Recommendation:** 4
**Confidence:** 3

**Summary:**

The paper introduces 3DPOV, a self-supervised post-training strategy designed to enhance the 3D spatial awareness of vision foundation models. While existing models excel at semantic tasks, they often fail to maintain multi-view consistency. To bridge this gap, 3DPOV leverages point tracking to enforce temporal alignment across patches. The core mechanism involves a teacher-student architecture where a student network learns to match the "similarity structure" of an anchor frame held by a teacher network. Instead of matching raw features, the model uses differentiable sorting to align the relative rankings of patches against a reference bank. Experiments on Probe3D benchmark demonstrate the superiority of the proposed method.

**Compliance With Llm Reviewing Policy:**

Affirmed.

**Key Questions For Authors:**

- How do you sample patch features from the batch? What is N_{ref} in Eq. (7)? Do you have multiple set of reference features for each query feature since Eq. (7) have subscript (t, r)?

- Why only finetune the last two layers of transformers during evaluation?

- What is the num of frames in each training video clip? Is it the number of frames mentioned in the ablation study? If so, why using such a short clip length? Can we increase the length of training clips?

**Limitations:**

yes

**Strengths And Weaknesses:**

## Strengths

- The rationale on computing loss on sorting instead of features is sound.
- The proposed method achieves outperforms existing methods by a large margin


## Weaknesses

- Reference Bank choice. The model uses a reference bank of patches sampled from other videos in the batch to establish relative similarity. If a training batch accidentally contains very similar videos, the "discriminative" power of the ranking decreases. Therefore, batch selection should be an interesting and important topic in the proposed framework. However, the authors did not provide analysis and ablation studies on this. Also, the size of each batch should also be disclosed and ablated, since batch size is an important factor that affects the self-supervised learning performance.

- Reliance on the point tracker. If the off-the-shelf point tracker makes an initial error in the anchor frame (frame 0), that error is "baked into" the teacher's signal. Because the student is trained to match the teacher's distribution, the model may learn to replicate the geometric hallucinations and result in errors.

- Some technical details need further clarification, see the questions section.

- Writing
  - Line 59 missing full stop, Line 64 framework spelling.
  - Page 8 is not complete, better move some content from appendix to the main paper.
  - Inconsistent reference format, some use conference name abbreviation while others do not.

---

> ### Author Rebuttal · Authors · 2026-03-31
>
> ### **1. Reference bank choice**
>
> We agree with the reviewer, the composition of the reference bank is indeed a key component which we have already addressed in Appendix Table 9. The mixture of internal and external frames in the reference pool has the aim of ensuring diversity in the similarity values (with internal references being patches sampled from frames of the current video processed, while external frames being patches sampled from frames of other videos in the batch). As such, for each entry in the batch, we sample 1 internal reference (excluding anchor frame) and 4 external references at random from frames within the batch.
>
> In order to further probe the reviewer's proposal, we kept the batch size consistent, but sampled external patches only from one other element from the batch. Results are reported in Table 7.
>
> _Table 7: SPair-71k Keypoint Matching; ablating the sampling of external references, we compare less discriminative sampling where external references are obtained from only one video in the batch_
> |Model|S / 0|M / 1|L / 2|All|
> |---|---|---|---|---|
> |DINOv2R|58.20|51.56|53.41|53.47|
> |Less discriminative sampling|59.91|52.41|54.31|54.95|
> |**3DPoV sampling**|**60.16**|**52.79**|**54.50**|**55.40**|
>
> The results strongly indicate that having discriminative references positively impacts the score . We will add the new ablation in the paper and continue to explore techniques to improve reference selection (ensure each external sample comes from a different video, compute similarity among frames and choose better candidates).
>
> The batch size used in all reported experiments is 32, we further add a breakdown of performance for varying batch sizes in Table 8. As shown, increasing batch size results in better scores across all viewpoints for 3DPoV.
>
> _Table 8: SPair-71k Keypoint Matching; batch size ablation_
> |Batch size|S / 0|M / 1|L / 2|All|
> |-----|---|---|---|---|
> |8|59.63|52.44|54.24|54.86|
> |16|59.77|52.40|54.32|54.95|
> |**32**|**60.16**|**52.79**|**54.50**|**55.40**|
>
> ---
> ### **2. Reliance on the point tracker**
> We start from a grid of points uniformly projected on the first frame of the processed video, as shown in Figure 2 and 7, not from an intermediate step. We apologise for the confusion and will clarify this.
>
> For visualization purposes, these entries show points only in masked areas, whereas in practice the grid will span across the entire frame (we will add more such examples in the camera-ready version). In that sense, the first frame cannot have any error, as each point in the grid is at full visibility over a pixel/visual location.
> Errors can arise only in subsequent frames, as showcased in Figure 6 and 3DPoV addressed such cases through using visible points and averaging over all tracked location during loss.
>
> ---
> ### **3. Clarify notation**
> $N_{ref}$ represents the number of reference features we aim to sample per one video entry. We comprise a pool of internal and external references per video entry, meaning that each entry in the batch gets a different set of references. In this sense, we refer to internal references as patches from the current video in the batch (excluding anchor frame), and external reference as patches from other videos in the batch (excluding current video frames). We will update the paper to include clearer notations and explanations
>
> ---
> ### **4. Finetuning the last two layers**
> Sorry for the misunderstanding, we will clarify this in the paper. We don't finetune during evaluation, we follow the protocol from Probe3D (frozen models for correspondence, only finetune DPT heads for the depth evaluations).
> For our training, we only finetune the last 2 layers as we found to be enough to achieve improvements. We leveraged the learned representations and, with minimal compute requirements we are able to improve 3D understanding. Table 12 in Appendix ablates number of unfrozen layers, repeated here.
>
> _Table 9: SPair-71k Keypoint Matching; we unfreeze a number of layers and experiment under the same setup_
> |Unfrozen Blocks|S / 0|M / 1|L / 2|All|
> |----|---|---|---|---|
> |Blocks 8-11 | 57.66| 49.84| 51.48| 52.50|
> |**Blocks 10-11**|**60.16**|**52.79**|**54.50**|**55.40**|
> |Blocks 11|58.64|51.72|53.41|53.79|
>
> ---
> ### **5. Lenght of video clips**
> The chosen setup currently uses 4 frames with step size 4, covering 12 frames, which we found enough to cover dynamics and viewpoint changes.
> Because our source datasets leverage rotations around an object/scene, more frames will not provide that much dynamic/visual information, and might actually lead to less informative correspondences, as relevant points become invisible (not weighted) due to rotation.
> The setup allows for using more frames, but due to compute limitations, this was the best configuration we could explore.
>
> ---
> ### **6. Fixing typos and formatting**
> Thank you for bringing this up to our attention. We will further polish the paper to resolve the typos, references and bring more content to the main paper.

---

> > ### Author Rebuttal · Reviewer_XHt8 · 2026-04-03
> >
> > I've read the authors rebuttal and would like to keep on weak accept rating. I believe the paper offers valuable contribution to the community.

---

> > > ### Author Response · Authors · 2026-04-05
> > >
> > > We are very glad that the additional experiments and clarifications were helpful in addressing your questions. Thank you for your thoughtful comments and for taking the time to engage with our rebuttal! Your comments directly guided several additions to the paper, improving its clarity and completeness.

---

### Official Review · Reviewer_ro4F · 2026-03-13

**Soundness:** 3
**Presentation:** 2
**Significance:** 3
**Originality:** 2
**Overall Recommendation:** 4
**Confidence:** 2

**Summary:**

The paper investigates how to improve 3D understanding in visual foundation models through a self-supervised post-training framework called 3DPoV, which enforces temporally consistent patch ordering across video frames. The authors highlight a key limitation of current models: while large-scale self-supervision yields strong 2D representations, these models still struggle with viewpoint-invariant spatial reasoning and multiview consistency. To address this gap, the paper integrates point tracking, a teacher-student framework, differentiable sorting, and a mixture of object-centric and scene-level video data to learn denser, more geometry-aware features. Experiments on Probe3D show consistent improvements in keypoint matching, depth estimation, and surface normal prediction across several backbone architectures.

**Compliance With Llm Reviewing Policy:**

Affirmed.

**Final Justification:**

Thank you to the authors for the detailed rebuttal and the additional analyses. My main concerns were largely addressed, especially through the expanded ablations on loss design, the clarification of comparisons with geometry-oriented models such as VGGT, and the discussion of why the gains on DINOv3 are smaller yet still meaningful in a highly saturated regime. I also found the broader discussion with other reviewers helpful: the added multi-seed results, batch-size and reference-bank ablations, and the follow-up clarifications on extreme-viewpoint evaluation and occlusion robustness make the empirical case substantially stronger. Overall, I view this paper as a technically solid and lightweight post-training approach for improving 3D-aware visual representations, with clear practical value despite some remaining limitations in evaluation breadth and presentation. On balance, I am supportive of a weak accept.

**Key Questions For Authors:**

In the main tables, the gains over DINOv3 are extremely small, sometimes nearly negligible. Do the authors believe this indicates that DINOv3 already internalizes much of the target invariance, or that the current post-training recipe is not yet fully matched to stronger backbones?

**Limitations:**

yes

**Strengths And Weaknesses:**

Strength
1. The paper addresses an important and well-defined gap in self-supervised visual representations by proposing a video-based post-training method that explicitly optimizes spatial consistency, rather than relying solely on temporal semantic propagation.
2. The method is evaluated on multiple Probe3D tasks, including keypoint matching, depth estimation, and surface normal prediction, and generally outperforms the baselines on both SPair and Navi. The improvements are particularly convincing under medium and large viewpoint changes.
3. The paper includes a thorough ablation study covering dataset mixture, number of frames, tracker choice, sorting strategy, anchor choice, and the number of unfrozen blocks. It also shows that the proposed post-training scheme is lightweight compared with large-scale pretraining.

Weakness
1. Although the overall trend is positive, the gains over DINOv3 are marginal in several tables. This raises the question of how significant the proposed method remains when applied to already strong, geometry-aware representations.
2. Some geometric foundation models, such as MASt3R and VGGT, also build on DINO-style vision encoders and produce tracking-related outputs. It would strengthen the paper to include comparisons with these methods as well.
3. The paper compares only two cross-entropy directions and attributes the observed difference to a mode-seeking effect, but it does not evaluate other plausible supervision objectives, such as symmetric losses, direct KL divergence, or temperature-controlled variants. As a result, the ablation supports the chosen direction only relative to one baseline, rather than within the broader space of reasonable loss designs.
4. There is a typo in line 65: “framework” is misspelled.

---

> ### Author Rebuttal · Authors · 2026-03-31
>
> ### **1. Significance of DinoV3 results**
> While the absolute gain is modest, it represents a meaningful improvement given the high saturation of DINOv3, which is distilled from a 7B-parameter teacher trained on data distributionally similar to NAVI. Crucially, this improvement is achieved with orders of magnitude less compute than the original pretraining (our finetuning amounts to only 0.028\% of a single DINOv3 epoch - 13 epochs of 9,242 samples using 4 frames comapred to 1689M) demonstrating that 3DPoV can efficiently extract geometric signal.
>
> Furthermore, these reported results relied on the DINOv2 hyperparameters;  we are currently optimizing the training regime specifically for DINOv3’s architecture and shall report the findings in the final version.
>
> ---
> ### **2. Comparison to geometric foundation models**
> VGGT is not a drop-in ViT encoder like DINO, so a 1-to-1 comparison is not straightforward. While DINO processes each image independently and directly outputs patch-aligned features, VGGT is a multi-view model that jointly parses image sequences, includes additional tokens and applies its own internal normalization.
>
> To approximate the DINO evaluation protocol, we enforced single-image inference, adjusted the input normalization (undoing the normalization probe3D does on SPair and allowing VGGT norm), and extracted only patch-aligned features. However, exact architectural parity is not possible due to VGGT’s inherently sequence-aware design. We nevertheless report results under this adapted setup in Table 5. A similar limitation applies to MASt3R, which operates on image pairs with joint geometric reasoning, making independent feature extraction non-native to its design. Nonetheless, we will add these works in the discussion and look into extending our comparison.
>
> _Table 5: SPair-71k Keypoint Matching; model comparison_
> |**Model**|**Backbone**|**S / 0**|**M / 1**|**L / 2**|**All**|
> |-----------|---|---|---|---|---|
> |DINOv2R|ViT-B/14|58.20|51.56|53.41|53.47|
> |**3DPoV**| DINOv2R-B/14|**60.16**|**52.79**|**54.50**|**55.40**|
> |VGGT|ViT-L/14|15.07|6.55|4.60|10.93|
>
>
>
> ---
> ### **3. Loss direction**
>
> We thank the reviewer for raising this point.  To address the reviewer concern we extend the new ablations including symmetric loss and KL divergence(variants) in Table 6. For KL we can write it as CE(t,s) - H(s). Also, we ablate different coefficients of H(s) to make the ablations even more comprehensive. As shown our default achieves SOTA, we attribute this to the chosen direction encouraging sharper target distributions, leading to more discriminative matching for correspondence tasks as explained in the paper.
>
> _Table 6: SPair-71k Keypoint Matching; ablation on loss direction_
> |**Loss**|**S / 0**|**M / 1**|**L / 2**|**All**|
> |----------------------------------|-----------|-----------|-----------|---------|
> |$CE(t,s)$|57.66|49.84|51.48|52.50|
> |**$CE(s,t)$**|**60.16**|**52.79**|**54.50**|**55.40**|
> |$CE(t,s) + CE(s,t)$|58.13|51.28|53.12|53.33|
> |$CE(t,s) - 0.1 * H(s)$|58.11|51.35|53.23|53.35|
> |$CE(t,s) - 0.8 * H(s)$|58.13|51.35|53.25|53.36|
>
> We believe the suggestions made by the reviewer are valuable to the discussion provided on this design choice, and thus aim to extend this ablation further for the camera-ready version.
>
> ---
> ### **4. Fixing typos**
> Thank you for bringing this up to our attention. We will further polish the paper to resolve the typos.

---

> > ### Author Rebuttal · Reviewer_ro4F · 2026-04-04
> >
> > Thank you to the authors for the careful rebuttal and for taking the time to address my comments in detail. I will consider to increase my original score.

---

> > > ### Author Response · Authors · 2026-04-05
> > >
> > > We are very glad that the additional experiments and clarifications were helpful in addressing your questions. Thank you for your thoughtful comments and for taking the time to engage with our rebuttal! Your constructive feedback and proposed ablations have helped improve the clarity and completeness of our work!

---

### Official Review · Reviewer_S6Cy · 2026-03-23

**Soundness:** 2
**Presentation:** 2
**Significance:** 2
**Originality:** 2
**Overall Recommendation:** 3
**Confidence:** 3

**Summary:**

This paper presents 3DPoV, a self-supervised post-training method for improving the 3D awareness of dense visual features from vision foundation models. The core idea is to leverage point tracks across video frames and enforce temporal consistency through a ranking-based loss: for each tracked patch, the method computes soft permutation matrices (via differentiable sorting) over a shared set of reference features, and trains a student network to match the teacher's anchor-frame rankings using a mode-seeking cross-entropy objective with visibility weighting. The method is evaluated on the Probe3D benchmark across keypoint matching (SPair-71k, Navi), depth estimation, and surface normal estimation, reporting improvements over DINO-family baselines (DINOv1, DINOv2R, DINOv3) as well as prior post-training methods such as NeCo, TimeTuning, and MoSiC.

**Compliance With Llm Reviewing Policy:**

Affirmed.

**Key Questions For Authors:**

See weaknesses.

**Limitations:**

There is no section named Limitations, but some potential limitations that come from the failure of the tracking model is covered in Section I.

**Strengths And Weaknesses:**

### **Strengths**
---
- **Principled extension of patch ordering to video supervision.** Extending ranking-based patch supervision from static images (NeCo) to video using point tracks is a meaningful methodological step. The use of temporal correspondences to enforce viewpoint-invariant similarity structure, rather than direct feature matching, is well-motivated and differentiates this work from simpler temporal consistency losses.
- **Coherent pipeline design.** The overall pipeline—teacher features at the anchor frame, student features at later frames, tracked patch extraction, reference patch bank, differentiable sorting, and permutation-based loss—is clearly laid out in Figure 1 and easy to follow. The mode-seeking CE direction with visibility weighting is a thoughtful design choice for encouraging confident rankings while handling occlusions.
- **Broad evaluation and thorough ablations.** The paper evaluates across multiple Probe3D tasks (correspondence, depth, normals), multiple backbones, and includes ablations on reference extraction, number of frames, dataset mixture, step size, resampling strategy, tracker choice, loss direction, and more. The ablation effort is more substantial than average and helps dissect the design.

### **Weaknesses**
---
- **Mismatch between training domain and evaluation domain.** The method trains on dynamic videos using point tracking estimates from CoTrackerV3, however, all evaluations are conducted on static 3D settings—Probe3D tasks measure feature consistency under viewpoint change for rigid, static scenes and objects. Based on the limited analysis on static scenes, it is difficult to understand the benefits of the proposed approach in dynamic scenarios. Essentially, if the user wants to build view-consistent features, there could be other approaches, such as using dense correspondence networks across viewpoints.

- **Limited comparison.** As mentioned in my previous weakness, the current evaluation is limited to static scenes, where other approaches of training view-consistent features become possible. Therefore, to verify the effectiveness of the proposed approach, the authors should include additional comparisions with works like "MULTIVIEW EQUIVARIANCE IMPROVES 3D CORRESPONDENCE UNDERSTANDING WITH MINIMAL FEATURE FINETUNING" (You et al. ICLR 2025) or "Improving 2D Feature Representations by 3D-Aware Fine-Tuning" (Yue et al. ECCV 2024).

- **Gains on DINOv3 are negligible, undermining the "backbone-agnostic" claim.** While the paper claims consistent improvements across a diverse set of backbones, including DINOv3, the actual performance improvement is modest. On SPair-71k, the DINOv3 improvement is 55.76 → 55.84 overall, with medium and large bins essentially unchanged. On Navi, the story is similar. This suggests the method's headroom shrinks significantly with stronger, more geometry-aware backbones. The current framing overstates the generality of the improvements—this claim should either be softened or supported with evidence of more substantial DINOv3 gains under different tuning configurations.

- **Limited statistical rigor for modest improvements.** No variance, confidence intervals, or multi-run statistics are reported anywhere across the main tables. With many gains in the range of 0.2–1.5 points, and some ablation differences as small as 0.3 points. Reporting mean ± std over at least 3 seeds for the main DINOv2R setting would significantly strengthen the paper.

- **Dependence on tracking quality is acknowledged but insufficiently analyzed.** The method relies entirely on CoTrackerV3 for supervision. While Figure 6 shows a failure case and visibility weighting is used to mitigate issues, there is no quantitative analysis of how often tracking errors occur, how severely they corrupt supervision, or how performance degrades on sequences with prolonged ambiguity or nonrigid motion. Since the entire training signal comes from these tracks, a more thorough robustness analysis would be expected.

- **The incremental benefit of differentiable sorting is modest.** Table 11 shows that removing the sorting module and applying the cross-entropy directly on similarity matrices only costs about 0.59 points overall on SPair-71k. Given the added complexity of the differentiable sorting pipeline, this raises questions about whether the sorting component is truly necessary or whether a simpler design could achieve comparable results.

Overall, based on the current weaknesses, I believe it is difficult to verify the advantages of the proposed training framework.

---

> ### Author Rebuttal · Authors · 2026-03-31
>
> ### **1. Mismatch between training domain and evaluation domain**
> As compared to correspondence based approaches (which typically require paired or synchronized multiview inputs at inference time),  we leverage videos as they provide natural viewpoint variation and implicit 3D structure. Using point tracking (e.g. CoTrackerV3), we obtain consistent correspondences over time, which lets the model learn view-invariant features without relying on explicit multiview supervision. For evaluation, we use Probe3D to ensure fair comparisons with correspondence based methods and show that videos provide a more natural way to leverage multi-view variations.
>
> We acknowledge that video based evaluation would further strengthen the paper and will look into extending in that direction!
>
> ---
> ### **2. Limited comparison**
> We report below (Table 1) the comparison with the proposed 3DCorrEnhance. While both methods improve on the baseline, 3DPoV achieves better scores across all viewpoint differences. We attribute it to the structured nature of the supervision in 3DPoV, which operates on relative relationships between patches rather than individual correspondences. This encourages more globally consistent feature organization, leading to consistent gains in keypoint matching performance.
>
> _Table 1: SPair-71k Keypoint Matching; Comparison to new baseline, both models are pretrained with DinoV2 with registers_
> |**Model**|**S / 0**|**M / 1**|**L / 2**|**All**|
> |------|---|---|---|---|
> |DINOv2R|58.20|51.56|53.41|53.47|
> |3DCorrEnhance|59.61|52.16|54.39|54.64|
> |**3DPoV**|**60.16**|**52.79**|**54.50**|**55.40**|
>
> ---
> ### **3. Significance of DinoV3 results**
> While the absolute gain is modest, it represents a meaningful improvement given the high saturation of DINOv3, which is distilled from a 7B-parameter teacher trained on data distributionally similar to NAVI. Crucially, this improvement is achieved with orders of magnitude less compute than the original pretraining (our finetuning amounts to only 0.028\% of a single DINOv3 epoch - 13 epochs of 9,242 samples using 4 frames comapred to 1689M) demonstrating that 3DPoV can efficiently extract geometric signal.
>
> Furthermore, these reported results relied on the DINOv2 hyperparameters;  we are currently optimizing the training regime specifically for DINOv3’s architecture and shall report the findings in the final version.
>
> ---
> ### **4. Variance among runs**
> We evaluate performance across three seeds in Table 2. As shown, the method has negligible variance resulting in reliable and consistent improvements over our baselines.
>
> _Table 2: Mean $\pm$ std across 3 seeds on SPair keypoint matching scores_
> |View diff|Mean|Std|
> |---|---|---|
> |0|60.17| 0.07|
> |1|52.71|0.09|
> |2|54.31|0.16|
> |0|55.29| 0.10|
>
> ---
> ### **5. Robustness of tracker module**
> 3DPoV is tracker agnostic, as shown in the ablation in Table 10 Appendix, which we put below again. Our method still improved the performance when leveraging even weaker point trackers such as RAFT, supporting that 3DPoV is tracker agnostic.
>
> _Table 3: SPair-71k keypoint matching; ablating choice of tracker_
> | **Tracker** | **S / 0** | **M / 1** | **L / 2** | **All** |
> |----|---|---|---|---|
> | DINOv2R Baseline| 58.20|51.56|53.41|53.47|
> | RAFT| 59.81|52.43|54.31|55.07|
> | CoTrackerV2|59.03|51.74|53.82|54.18 |
> | **CoTrackerV3**| **60.16**| **52.79**| **54.50**| **55.40**|
>
> This means our method can benefit even further from better trackers according to the field progress, in a plug-and-play manner.
>
> In a qualitative manner, on the datasets used it was particularly hard to find failure cases (example shown in Figure 6 showcases failure for challenging scenarios), which further supports current performance of the chosen tracker.
> Nonetheless, tracking failure is possible. While we can't directly evaluate tracking errors on our datasets due to lack of ground truth, we refer to the comprehensive analysis done in the original CoTracker V3 paper (Nikita et. al.) study, highlighting Table 2.
>
> ---
> ### **6. The benefit of the differentiable sorting component**
> While directly leveraging similarity matrices showcases a degradation in SPair scores, we observe an even grater drop for the Navi eval in Table 4.
>
> _Table 4: Navi Correspondance._
> |Method|$\theta_{0}^{15}$|$\theta_{15}^{30}$|$\theta_{30}^{60}$|$\theta_{60}^{120}$|
> |----|---|---|---|---|
> | DINOv2R Baseline|87.92|67.74|47.18|**31.57**|
> |Similarity matrix|88.70|68.42|47.03|31.21|
> |**Sorted similarity matrix**|**89.22**|**69.23**|**47.48**|31.33|
>
> Without a sorting component, the method struggles to perform, achieving better results in easier viewpoint cases, but degrading baseline score across the last two buckets. The DiffSort modules allows for a more  complex mapping, that prioritizes high similarity matches while also weighting in possible alternatives, which proves efficient in more challenging viewpoint scenarios. We thank the reviewer for raising this point and will add it to the discussion.

---

> > ### Author Rebuttal · Reviewer_S6Cy · 2026-04-04
> >
> > I thank the authors for the rebuttal. May the authors provide additional explanation on how the provided table 2 is obtained?

---

> > > ### Author Response · Authors · 2026-04-04
> > >
> > > # 1. Variance among runs
> > >
> > > Our code is deterministic under a fixed training seed. We set the seed to 3 different values: 0, 42 and 100 and train 3 different models. Then, we evaluate these models on SPair keypoint matching and report mean and standard deviation per viewpoint bucket. All experiments are conducted with DINOv2-reg, vit14 variant of 3DPoV, with the same setup as explained in Appendix D.
> > >
> > > We repeat the table here, there was a typo in the last row label, where the category is "All" not 0, matching SPair buckets. We apologize for this.
> > >
> > > _Table 2: Mean $\pm$ std across 3 seeds on SPair keypoint matching scores_
> > > |View diff|Mean|Std|
> > > |--|--|--|
> > > |S/0|60.17| 0.07|
> > > |M/1|52.71|0.09|
> > > |L/2|54.31|0.16|
> > > |All|55.29| 0.10|
> > >
> > > We further evaluate variance across these checkpoints on NAVI correspondence in Table 16. The low variance on both NAVI and SPAIR datasets further supports the stability of our training pipeline and the consistency of our gains. We are happy to include additional experiments if further clarification is needed.
> > >
> > > _Table 16: Mean $\pm$ std across 3 seeds on NAVI correspondence_
> > > |View diff|Mean|Std|
> > > |--|--|--|
> > > |$\theta_{0}^{15}$|89.14|0.11|
> > > |$\theta_{15}^{30}$|68.85|0.17|
> > > |$\theta_{30}^{60}$|47.61|0.03|
> > > |$\theta_{60}^{180}$|31.65|0.02 |

---

### Decision · Program_Chairs · 2026-04-30

**Decision:**

Accept (regular)

**Comment:**

The paper introduces 3DPOV, a self-supervised post-training strategy designed to enhance the 3D spatial awareness of vision foundation models. The method leverages point tracking across video frames to enforce temporal nearest-neighbor consistency through a ranking-based differentiable sorting loss.

The reviewers' consensus leaned toward acceptance (3 Weak Accepts, 1 Weak Reject who indicated willingness to raise their score). Reviewers praised the methodological elegance of the sorting module, the robust 3D emergence achieved with minimal computational cost, and the thorough multi-task evaluations on the Probe3D benchmark. Reviewers initially raised critical concerns about a performance regression at extreme viewpoint shifts, the necessity of the sorting module, and the lack of multi-seed statistical variance reporting.

During the rebuttal phase, the authors addressed these concerns effectively. Most notably, they discovered a center-padding bug in the upstream Probe3D evaluation codebase that disproportionately penalized extreme viewpoints for ViT-14 models. Once patched, the architecture and sorting module demonstrated uniform improvements across all viewpoint geometries. The authors also provided extensive multi-seed variance tables proving training stability. The reviewers independently verified the upstream bug, found the new data convincing, and officially acknowledged that their concerns were fully resolved.

The paper is technically solid, well-motivated, and provides a lightweight, practical approach to improving 3D-aware visual representations. Therefore, we recommend acceptance. Please ensure that the final camera-ready version incorporates all the changes promised during the rebuttal phase.